# Neural energy coding patterns of dopaminergic neural microcircuit and its impairment in major depressive disorder: A computational study

Yuanxi Li[1]*[☯], Bing Zhang[2,3][☯], Jinqi Liu[4], Rubin Wang[1]*

**1** Institute for Cognitive Neurodynamics, School of Mathematics, East China University of Science and Technology, Shanghai, China, **2** Department of Anesthesiology, Obstetrics and Gynecology Hospital of Fudan University, Fudan University, Shanghai, China, **3** Department of Anesthesiology, Shanghai Key Laboratory of Maternal Fetal Medicine, Shanghai Institute of Maternal-Fetal Medicine and Gynecologic Oncology, Clinical and Translational Research Center, Shanghai First Maternity and Infant Hospital, Tongji University School of Medicine, Shanghai, China, **4** University of Rochester, Rochester, New York, United States of America

☯ These authors contributed equally to the manuscript
* dr.yuanxili@gmail.com (Y.L.); rbwang@ecust.edu.cn (R.W.)

## Abstract

Numerous experiments have found that the behavioral characteristics of major depressive disorder (MDD) animals are usually associated with abnormal neural activity patterns and brain energy metabolism. However, the relationship among the behavioral characteristics, neural activity patterns and brain energy metabolism remains unknown. In this paper, we computationally investigated this relationship, with a particular focus on how neural energy coding patterns change in MDD brains, in the VTA-NAc-mPFC dopaminergic pathway of the reward system based on our biological neural network model and neural energy calculation model. Interestingly, our results suggested that the neural energy consumption of the whole VTA-NAc-mPFC microcircuit in MDD group was significantly reduced, which was mainly attributed to the decreasing neural energy consumption in the mPFC region. This observation theoretically supported the view of low-level energy consumption in MDD. We also investigated the neural energy consumption patterns of various neuronal types in our VTA-NAc-mPFC microcircuit under the influence of different dopamine concentrations, and found that there were some specific impairments in MDD, which provided some potential biomarkers for MDD diagnosis. More specifically, we found that the actual neural energy consumption of medium spiny neurons (MSNs) in the NAc region was increased in the MDD group, whereas pyramidal neurons in the mPFC region exhibited higher actual neural energy consumption in the NC group. Additionally, in both neuron types, the actual neural energy required to generate an action potential was higher in the MDD group, suggesting that, given the same energy budget, these neurons in the MDD group tended to generate fewer action potentials. To further explore the relationship between neural coding patterns and neural energy coding patterns in the VTA-NAc-mPFC microcircuit, we in addition calculated P-V correlation for each neuronal type, defined as the Pearson's correlation coefficient between membrane potential and neural power. The results showed

**Data availability statement:** The MATLAB codes that support the findings of this study are available at GITHUB (https://github.com/Yuanxi-Li/MDD-NeuralEnergy).

**Funding:** This study was supported by the National Natural Science Foundation of China (11872180 and 12072113 to R.W.; 82271582 to B.Z.) and the Biomedical Engineering Special Program of 'Science and Technology Innovation Projects' from Shanghai Science and Technology Commission (23S11900600 to B.Z.). The funders had no role in study design, data collection and analysis, decision to publish, or preparation of the manuscript. None of the authors received the salary from these funding sources.

**Competing interests:** The authors have declared that no competing interests exist.

that the membrane potential and neural power were not perfectly correlated (P-V correlations ranged from 0.6 to 0.9), and dopamine concentration inputs affected the P-V correlations of the MSN, pyramidal neurons and CB interneurons in the mPFC region. These findings suggested that the joint application of the neural coding theory and neural energy coding theory will be superior to the application of any single theory, and this joint application could help discover new mechanisms in neurocircuits of MDD. Overall, our study not only uncovered the neural energy coding patterns for the VTA-NAc-mPFC neural microcircuit, but also presented a novel pipeline for the study of MDD based on the neural coding theory and neural energy coding theory.

## Author summary

Major depressive disorder (MDD) disrupts the brain's reward system, leading to impaired value judgment and decision-making based on environmental cues and individual experiences. Since decision-making processes are directly encoded by neural activity, which is associated with brain energy consumption, understanding the relationship between neural activity and neural energy consumption is crucial for explaining the abnormal brain function in MDD. In this study, we used computational models to investigate how neural energy consumption patterns change in MDD. We mainly focused on the VTA-NAc-mPFC dopaminergic pathway, which is a key neural circuit in the reward system. Our findings revealed that actual neural energy consumption of this pathway was lower in MDD, primarily due to reduced energy use in the prefrontal cortex. We also identified distinct energy consumption patterns among different neuron types within this circuit. These insights suggest that neural energy patterns could serve as potential biomarkers for MDD and highlight the importance of integrating neural coding and neural energy coding theories. Our study provides a novel perspective on MDD, offering new insights into how it affects brain function.

## Introduction

The most significant behavioral characteristic of mice with major depressive disorder (MDD) is their inability to make correct value judgement and decisions based on the environment information and their experiences [1,2]. Specifically, mice with MDD usually show a lack of interest in rewards, and on the other hand, they are unable to properly avoid punishments [1,2]. Previous observations have exhibited that the function of the reward circuitry in the MDD brains, especially of the most critical dopaminergic neural pathway as the ventral tegmental area (VTA) - nucleus accumbens (NAc) - medial prefrontal cortex (mPFC) dopaminergic pathway, is impaired [1–4]. Recently, evidence from both experimental and computational modeling studies has suggested that the abnormally reduced dopamine levels in this neural pathway is the key mechanism underlying the anomalous reward behavior and electrophysiological characteristics in MDD [5–8].

According to the neural coding theory [9], mice with MDD are inevitably associated with different neural activity patterns and encoding strategies in their brains [6,8]. Since the encoding of neural activity is always accompanied by the neural energy consumption [10,11], MDD has distinct neural energy coding (consisting of energy supply and energy consumption) patterns, which have been observed in abundant studies: 1) At the molecular level, multi-omics

results have exhibited differences in the levels of energy-metabolism-related metabolites in the cerebrospinal fluid of MDD animals and human compared to normal control (NC) group [12,13]; 2) At the cellular level, the upregulated astroglial Kir4.1 channel leads to abnormal neuronal bursts in depression [14], while astrocytes are also responsible for supplying energy to neurons, suggesting a possible difference in their neural energy coding patterns; and 3) At the large-scale brain network level, functional magnetic resonance imaging (fMRI) experiments have demonstrated different blood oxygen level-dependent (BOLD) signals and functional connectivity among regions in MDD brains [15]. However, the relationship between the abnormal neural activity patterns and the neural energy coding patterns in VTA-NAc-mPFC pathway has not been uncovered yet [8,16–18]. Additionally, there is a lack of theoretical consideration on the underlying mechanism of neural energy coding patterns. It is well-known that the brain follows the efficient energy coding principles including energy minimization and efficiency maximization [19], but it remains unclear whether the VTA-NAc-mPFC pathway with MDD adheres this principle.

To the best of our knowledge, we were the first to establish a neurodynamical model [8] and neural energy computation model [17] for the mouse VTA-NAc-mPFC neural microcircuit, and tried to address these questions previously. We quantitatively investigated the differences in neural coding patterns and the energy consumption by ion channels (ionic neural energy) between MDD and NC groups, and especially, explained how ion channel kinetics and excitation-inhibition (E-I) balance in MDD group dynamically changed these patterns [17]. More specifically, we mathematically identified four important ion channels (the KAs sodium ion channel in the medium spiny neurons of the NAc region, the KS sodium ion channel and the NaP potassium ion channel in the proximal dendrite of pyramidal neuron in the mPFC region, and the Ca calcium ion channel in the soma of pyramidal neuron in the mPFC region) and the E-I balance in the pyramidal neurons of the mPFC region as critical factors influencing circuit neurodynamics between the MDD and NC groups [8]. Our neural energy calculation further revealed that the ionic neural energy consumption of the whole VTA-NAc-mPFC network in MDD group was lower than that in NC group [17]. However, despite the above progress, three important questions are still unanswered: 1) Is there any difference in the neural energy efficiency between the MDD and NC groups when encoding neural activities? 2) Is there any specific pattern or biomarker related to neural energy that can be used to diagnose MDD? and 3) What is the correlation between neuronal activity (i.e., the membrane potential) and its instantaneous neural energy (neural power), and does MDD alter this correlation?

In this paper, we further addressed the above three questions based on our VTA-NAc-mPFC neural microcircuit and neural energy models [8,17]. We investigated the neural energy coding patterns and energy efficiency under four different dopamine concentration input situations (Low, Medium, High, and Full. See **Methods** and S1 File). Interestingly, our simulation results showed that for the two most important types of neurons in this microcircuit, the pyramidal neurons (Pyra) in the mPFC region and medium spiny neuron (MSN) in the NAc region, more neural energy was required to encode a single action potential when we decreased the dopamine concentration input, suggesting that these neuronal types encoded neural information less efficiently under the MDD situation. The result also exhibited that the neural energy consumption of the whole VTA-NAc-mPFC microcircuit in MDD group was significantly reduced, which was mainly attributed to the decreasing neural energy consumption in the mPFC region, but not the NAc region. Significantly, this observation supported the view of low-level energy consumption in MDD [20–22], and also suggested the potential of neural energy coding patterns as biomarkers for MDD diagnosis. Additionally, for each neuronal type, we calculated the P-V correlations, described as the Pearson correlation

coefficients between the membrane potential curve and the corresponding neural power curve, under different dopamine concentration inputs. The results showed that the membrane potential curve and the associated neural power curve were not perfectly correlated (ranging from 0.6 to 0.9), and the correlations of the MSN, the Pyra neurons, and the CB interneurons of the mPFC region in MDD group were significantly changed compared with those of the NC group. These findings suggest that integrating neural coding theory with neural energy coding theory is valuable for the study of our VTA-NAc-mPFC microcircuit and MDD. Overall, our study not only uncovered the neural energy coding patterns for the VTA-NAc-mPFC neural microcircuit, but also presented a novel framework for studying MDD through the integration of neural coding and neural energy coding theories.

## Methods

### Biological neural network modeling of the VTA-NAc-mPFC neural microcircuit

The key dopaminergic neural pathway in the reward system [4,8], including the VTA, the NAc, and the mPFC regions, was studied in this paper. To model the neural activities of this pathway, we built the VTA-NAc-mPFC neural microcircuit neurodynamical model based on the mouse brain atlas and our previous work (See S1 File for model details) [4,8]. Briefly, our VTA-NAc-mPFC neural microcircuit model consisted of 28 coupled neurons from the above three regions (Fig 1a). Among them, the NAc region consisted of one medium spiny neuron (MSN, D2-type), one parvalbumin interneuron (PV interneuron), and one calbindin interneuron (CB interneuron), where the morphology of MSN in the model was simplified as the structure consisting of one soma and ten identical dendrites (Fig 1b), and the morphology of the interneuron was simplified as a point neuron including only a soma. The mPFC region consisted of twenty pyramidal neurons (Pyra neurons, D1-type), three PV interneurons, and two CB interneurons, where the morphology of Pyra neuron was simplified as the structure consisting of one soma, one proximal dendrite, and one distal dendrite (Fig 1c).

The VTA region mainly consisted of dopaminergic (DA) neurons, but DA neurons were not added to our model due to insufficient experimental evidence [8]. Alternatively, we set up a global dopamine concentration parameter, categorized into four levels (Low, 0–25%; Medium, 25%-50%; High, 50%-75%; and Full, 75%-100%. Fig 1d). For each case, dopamine input at each moment followed a uniform distribution within its range, in order to reflect the dynamic neuronal activity of DA neurons and to simulate the dynamic dopamine concentration inputs to the NAc region and the mPFC region. According to the physiological experimental evidence [23–27], the kinetics of some ion channels in the NAc and the mPFC regions were mediated by different dopamine concentration inputs, including the Ca channel, the Can channel, the KS channel, and the NaP channel of the Pyra neurons in the mPFC region, and the KAs channel, CaL1.2 channel of the MSN in the NAc region. Thus, the changes in kinetics led to distinct neuronal firing dynamics and network activity dynamics [9]. We modeled this effect in our VTA-NAc-mPFC microcircuit model (See S1 File for details) [8].

The synaptic projections and their kinetics in our VTA-NAc-mPFC microcircuit were modeled based on the chemical synapse models [28], which contained the AMPA and NMDA excitatory synapse types, and the $GABA_a$ inhibitory synapse type. The compartmental neuron models and the compartment currents [9,29] were incorporated to model the electrical behavior of dendrites in MSN and Pyra neurons. In addition, each neuron in the VTA-NAc-mPFC microcircuit was stimulated by an external stimulus current.

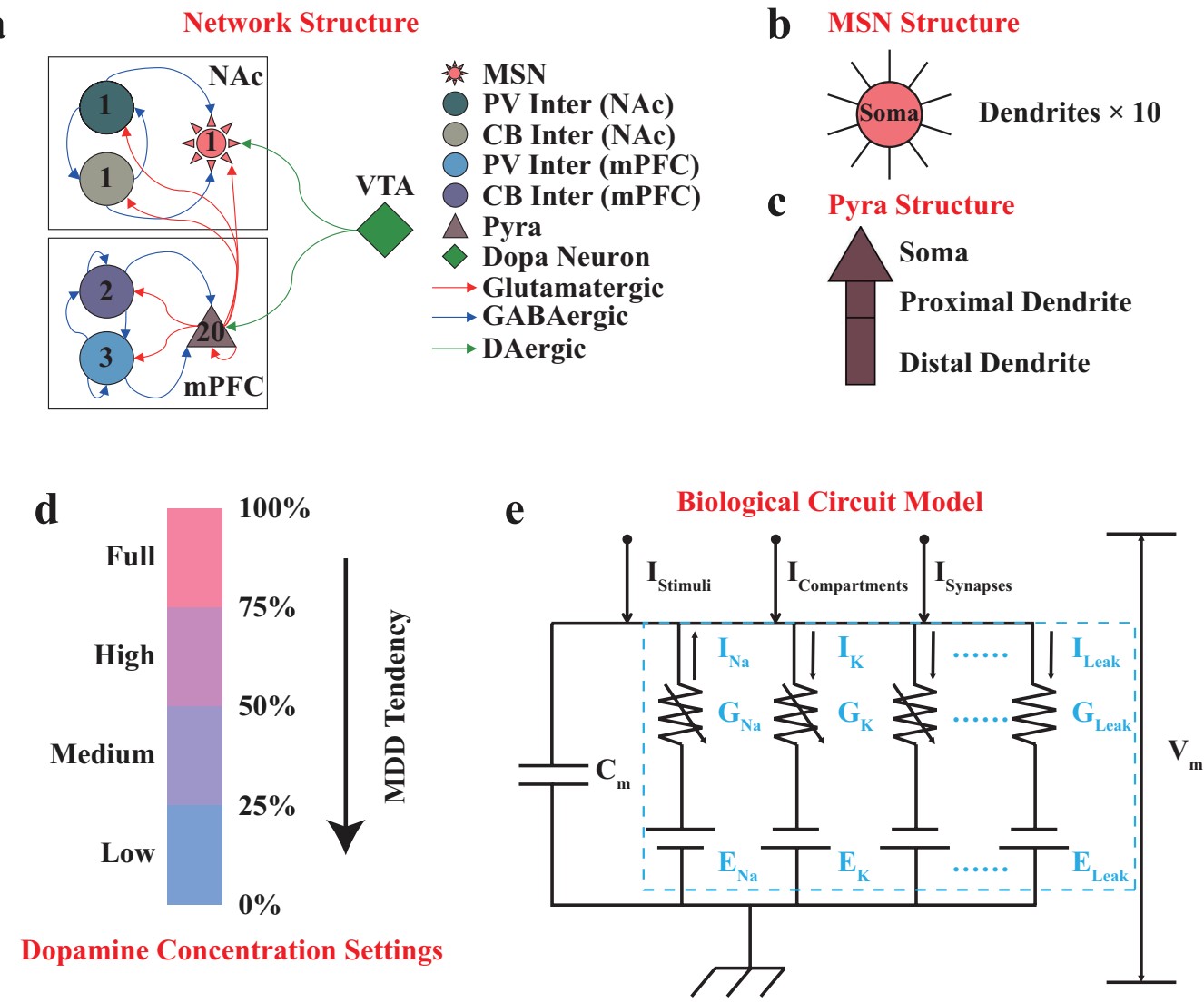

**Fig 1. The schematic diagrams of the VTA-NAc-mPFC neural microcircuit model and the H-H model.** (a), topological structure of the VTA-NAc-mPFC neural microcircuit; the numbers inside denoted the numbers of this neuronal type that we modeled in the microcircuit, the arrows of different colors denoted different neurotransmitter receptors, the directions denoted the directionality of synaptic connections, different types of neurons were shown as different shapes and colors. There were 1 MSN, 1 PV interneuron and 1 CB interneuron in the NAc region, while 20 Pyra neurons, 3 PV interneurons and 2 CB interneurons in the mPFC region. (b), the morphologic structure of the MSN, which consisted of 1 soma and 10 identical dendrites. (c), the morphologic structure of the pyramidal neuron in mPFC, which consisted of 1 soma, 1 proximal dendrite and 1 distal dendrite. (d), the settings of dopamine concentration in the model. (e), the schematic diagram of the H-H model, where the ionic currents were considered as the internal currents (labeled with light blue), and the stimulus, synaptic, and compartment currents were considered as the external currents. The figure was reproduced from Li *et al.* [8,17] with the authors' permissions.

We used the Hodgkin-Huxley (H-H) model [30] to simulate the membrane potential for each individual neuron in our VTA-NAc-mPFC microcircuit, where different types of neurons in the above brain regions were equivalent to distinct electronic circuit structures [8]. The basic form of our Hodgkin-Huxley (H-H) model can be described as Equation (1), which contains four types of currents in the equivalent circuit (Fig 1e): the ion channel currents $I_{Ions}$, the compartment currents $I_{Compartments}$, the synaptic currents $I_{Synapses}$, and the stimulus currents $I_{Stimuli}$.

$$C_m \frac{dV_m}{dt} + \sum I_{Ions} + \sum I_{Compartments} + \sum I_{Synapses} = I_{Stimuli} \tag{1}$$

The complete VTA–NAc–mPFC microcircuit model [8] consisted of 69 H-H equations as shown in Equation (1), which represented a set of ordinary differential equations (ODEs) with 1,527 variables to be solved. Since the ion channel kinetics and the neurotransmitter-receptor binding kinetics were diverse, the neural microcircuit had complex dynamics. The detailed modeling and the related parameters can be found in S1 File and our previous works [8].

Based on these models, we previously studied the network activities and neurodynamics in the VTA-NAc-mPFC microcircuit, and we built the link between lower dopamine concentration inputs and more intense depressive-like status [8]. Briefly, it has been found that the NAc region and the mPFC region had abnormally low levels of dopamine concentration and metabolic rates [3,5,6], with reduced firing rates and burst frequency of the Pyra neurons in the mPFC region [7,26], and no significant change in the firing rates of the MSNs in the NAc region [31]; By setting gradually lower levels of dopamine concentration input in our model, our simulation results were gradually in line with these depressive-like electrophysiological features [8]; The abnormal electrophysiological features in MDD gradually disappeared as we increased the dopamine concentration inputs from Low to Full [8]. Thus, the 'Low' level of dopamine concentration was considered as the MDD group, while the 'Full' level of dopamine concentration was considered as the NC group. The 'Medium' and 'High' dopamine levels were considered intermediate states between MDD and NC, representing conditions biased toward the NC state and the MDD state, respectively.

## Neural energy calculations based on the VTA-NAc-mPFC neural microcircuit model

Since the neural activities are modeled as the equivalent electrical circuit activities based on the H-H model, the energy consumed by neural activities, i.e., neural energy, can be described as the energy consumed in the H-H electrical circuit [17]. According to our previous neural energy model and the Moujahid's neural energy model [17,32], the instantaneous neural energy (neural power) corresponding to Equation (1) can be described as Equation (2).

$$H_{All} = H_{Stimuli} + \sum H_{Compartment} + \sum H_{Synapses} - \sum H_{Ions} \tag{2}$$

Where, $H_{All}$ denoted the neural power of the whole H-H circuit, which was the vector sum of the four power components (the ion channel power $H_{Ions}$, the compartment power $H_{Compartments}$, the synaptic power $H_{Synapses}$, and the stimulus power $H_{Stimuli}$) corresponding to the current types. As described in our previous study [17], although the AMPA and NMDA receptor-mediated excitatory postsynaptic currents (EPSCs) and excitatory postsynaptic potentials (EPSPs) are induced by the opening of cationic channels, and the GABA$_a$ receptor-mediated inhibitory postsynaptic currents (IPSCs) and inhibitory postsynaptic potentials (IPSPs) are induced by the opening of chloride channels, the synaptic currents in our model were considered as the external currents based on the generally accepted chemical synapse computational models in this field [28]. The stimuli currents and the compartment currents were also considered as external currents [17]. Accordingly, these power components can be calculated by Equation (3). The detailed model description and parameter settings can be found in S1 File and our previous papers [17]. Then, the neural energy can be calculated by integrating the power over the time, with a simulation duration of 3,000 ms in our study.

$$\begin{cases} H_{\text{Stimuli}} = V_m I_{\text{Stimuli}} \\ \sum H_{\text{Compartment}} = V_m \sum I_{\text{Compartment}} \\ \sum H_{\text{Synapse}} = V_m \sum I_{\text{Synapse}} = \left( \sum I_{\text{AMPA}} + \sum I_{\text{NMDA}} + \sum I_{\text{GABA}} \right) V_m \\ \sum H_{\text{ions}} = \sum I_{\text{ions}} \left( V_m - E_{\text{ions}} \right) \end{cases} \quad (3)$$

Previously, we have focused on the energy consumption by ion channels, i.e., $\sum H_{\text{ions}}$ in Equation (3) (labeled with light blue in Fig 1e and referred to as 'ionic neural energy' or 'ionic energy'), for each neuronal type in our VTA-NAc-mPFC microcircuit [17]. The ionic neural energy represented the theoretical energy consumption required by an individual neuron to encode its neural activities. In other words, it measured the total amount of energy required to move the ions across the cell membrane through ion channels (including passive and active transport). The simulation results have shown that the ion channels consumed neural energy all the time, especially consuming much more energy during the generation of action potentials [17].

However, according to Equation (3), the synaptic currents, stimulus currents, and compartment currents may consume or supply neural energy (Fig 2). Here, we let the positive neural energy and neural power denote the energy consumption, and negative neural energy

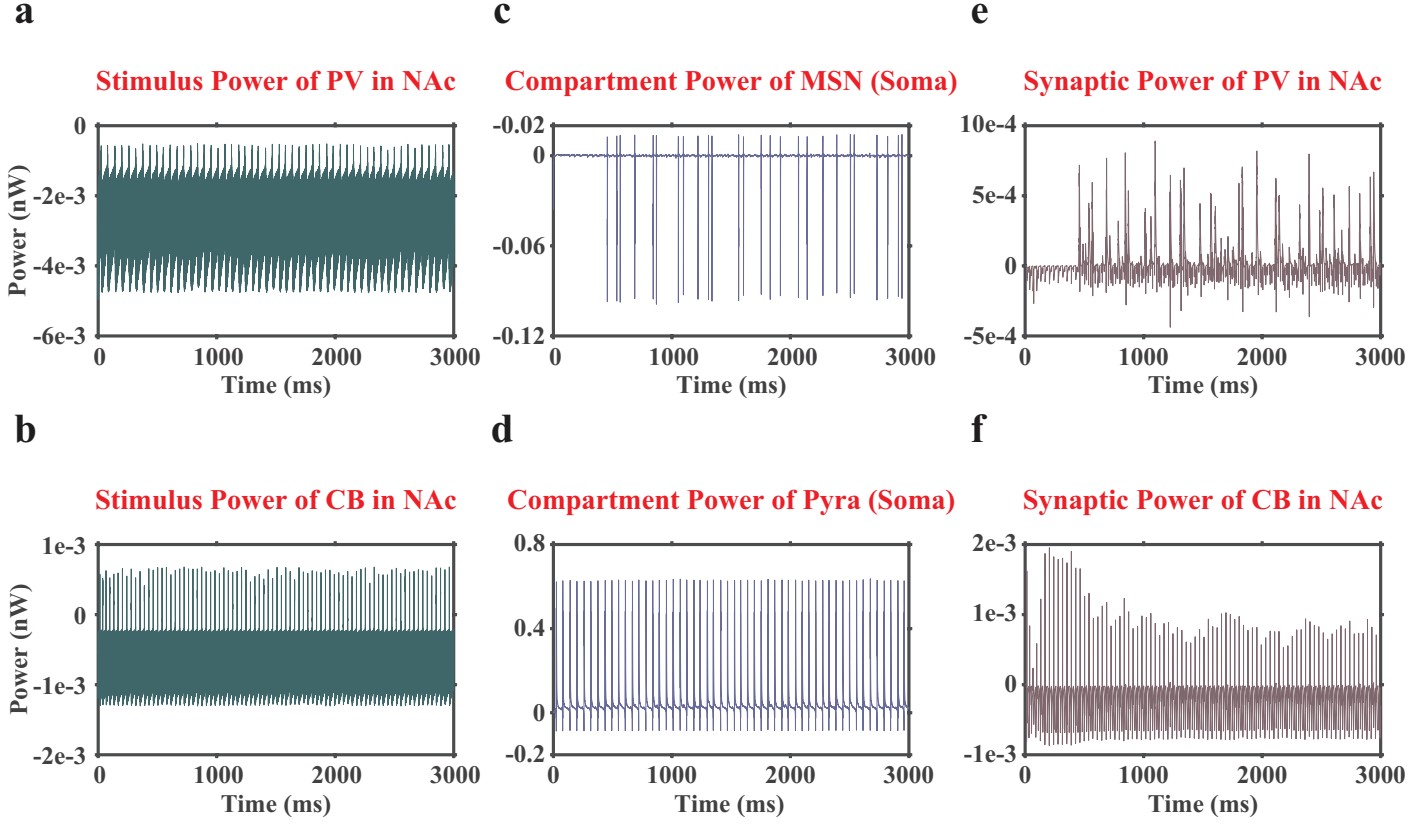

**Fig 2. Representative images of neural powers associated with stimulus currents, compartment currents and synaptic currents (only displaying one replicate under Low dopamine concentration).** (a), the neural power associated with the stimulus currents of the PV interneuron in the NAc region. (b), the neural power associated with the stimulus currents of the CB interneuron in the NAc region. (c), the neural power associated with the compartment currents of the MSN soma in the NAc region. (d), the neural power associated with the compartment currents of the soma of Pyra neurons in the mPFC region. (e), the neural power associated with the synaptic currents of the PV interneuron in the NAc region. (f), the neural power associated with the synaptic currents of the CB interneuron in the NAc region.

and neural power denote the energy supply. The neural energy consumption of the H-H equivalent electrical circuit $H_{All}$ can be calculated by Equation (2). To distinguish it from 'ionic (neural) energy' [17], we referred to it as 'actual (neural) energy', which reflected the actual neural energy consumption of a neuron in the VTA-NAc-mPFC neural microcircuit.

## Model application and simulation

In this paper, we calculated the actual energy consumption $H_{All}$ in our VTA-NAc-mPFC microcircuit, and investigated its pattern changes under different dopamine concentration inputs. The simulations were mainly performed on Windows 10 (version: 1511, CPU: i5-6400, RAM: 32 GB) using MATLAB (R2022a, MathWorks, https://www.mathworks.com/).

We mainly focused on two features for the actual energy consumption: 1) The amount during 3,000 ms; and 2) The amount for generating per action potential. Together with our previous findings on the membrane potential patterns and ionic energy consumption patterns [8,17], we then summarized the neural energy coding patterns of MDD, and established a novel pipeline based on the neural coding theory and neural energy coding theory.

## Statistics and reproducibility

In order to get robust conclusions, especially to avoid the influence of the stochastic variables, 9 replicates for each dopamine concentration input (Low, Medium, High, and Full) were conducted (See S1 File for details). The simulated results were statistically analyzed to ensure the significances of the findings.

All statistical tests were two-tailed, and significance was assigned at $P<0.05$. Normality was assessed by Shapiro–Wilk test. Equal variances between group samples were assessed by Brown–Forsythe test. One-way ANOVA with Tukey's multiple comparison test was used for comparing the differences among multiple groups when the data followed normal distributions and no differences among the variances, Brown–Forsythe and Welch ANOVA tests with Games-Howell multiple comparison test were used when the data followed normal distributions with differences among the variances, and Kruskal–Wallis test with Dunn's multiple comparison test was used if the data were non-normally distributed. Unless otherwise noted, the reported statistics were all presented as the mean values with the standard error of the mean (means ± SEM). All the statistics were performed in GraphPad 8.0 (GraphPad Prism, GraphPad Software, www.graphpad.com).

Note that there were 20 Pyra neurons, 3 PV interneurons and 2 CB interneurons in the mPFC region. Due to the stochastic variables, even the same type of neurons may exhibit different patterns. To avoid this bias, we averaged the results for statistical analysis.

## Results

### Significantly negative correlation between dopamine concentration inputs and actual energy consumption of MSN

Previous computational and experimental studies have shown that there is no significant difference in the membrane potential encoding patterns of MSNs in the NAc region at different dopamine concentrations [8,16,31], but the ionic energy consumption of the whole MSN (consisting of one soma and ten dendrites) in MDD group during 3,000 ms was significantly lower than that in NC group (MDD=0.223 nJ, NC=0.256 nJ; reported here as the average of 9 replicates) [17].

In this paper, we reported that both the actual energy consumption during 3,000 ms and per spike of MSN were negatively correlated with dopamine concentrations (Figs 3a-c and

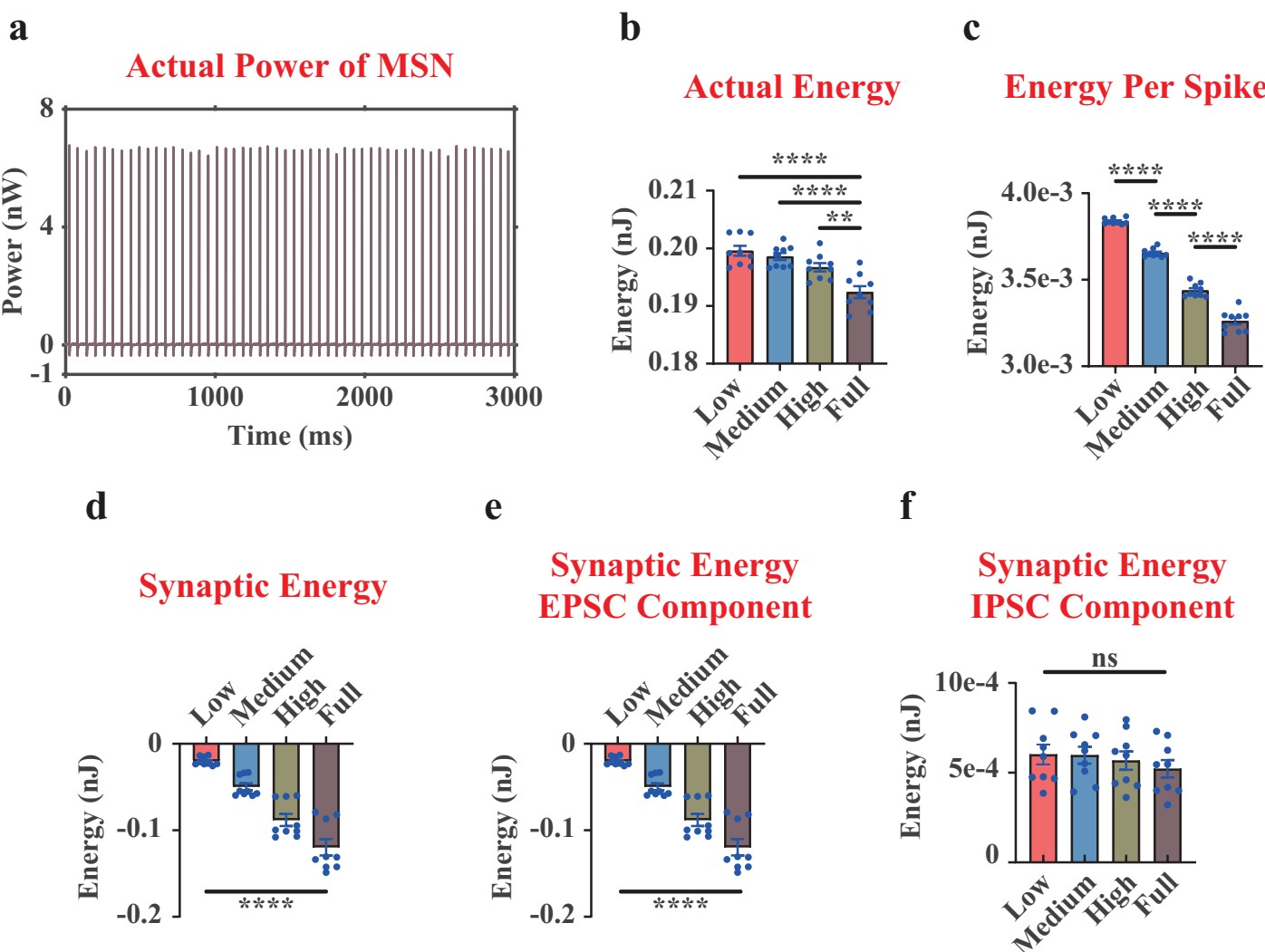

**Fig 3. The actual neural power and actual neural energy results of the whole MSN (including one soma and 10 identical dendrites).** (a), representative curves of the actual power of the whole MSN (only displaying one replicate under Low dopamine concentration). (b), quantifications of the actual neural energy consumption of the whole MSN during 3,000 ms. (c), quantifications of the actual neural energy consumption per spike of the whole MSN. (d), quantifications of the neural energy of the synaptic current component of the whole MSN. (e), quantifications of the neural energy of the EPSC component of the whole MSN. (f), quantifications of the neural energy of the IPSC component of the whole MSN. n=9. ns, non-significant. **, p<0.01. ****, p<0.0001.

S1a). In contrast with the ionic energy consumption patterns of MSN [17], the actual energy consumption in MDD group was significantly higher than that in NC group (MDD=0.200 nJ, NC=0.192 nJ; Fig 3b), suggesting that the combined effect of three external currents (stimulus, synaptic, and compartment) supplied more neural energy in the NC group (MDD=-0.023 nJ, NC=-0.064 nJ; The negative sign denoted the energy supply).

Since the impairment of E-I balance has been found to be one of the key mechanisms of MDD [8,17,33], we investigated the neural energy component associated with synaptic currents in MSN. The results exhibited a significantly higher energy supply by synaptic currents in NC group (MDD=-0.020 nJ, NC=-0.120 nJ; Fig 3d). Further results indicated that the cause of it was an increase in energy supplied by EPSCs (Fig 3e) rather than a decrease in energy consumed by IPSCs (Fig 3f) when the dopamine concentration rose, demonstrating from

the neural energy coding perspective that EPSCs in E-I balance of MSN were more sensitive to dopamine concentration inputs. This observation was also in line with the fact that more dopamine inputs will dramatically increase the firing rates of the Pyra neurons in mPFC [6–8,23,26], resulting in the higher glutamate level in the VTA-NAc-mPFC microcircuit.

### Lower actual energy consumption with higher actual energy consumption per spike in Pyra neurons with MDD

Pyra neuron in the mPFC is one of the most important excitatory neuronal type that encodes the social and cognitive behaviors [23,26,27]. There were 20 Pyra neurons (consisting of one soma, one proximal dendrite, and one distal dendrite) in the mPFC region in our microcircuit model, and the most important role of Pyra neurons in the network was to release glutamate neurotransmitters and excite the microcircuit [8]. Previous evidence suggested that the firing rate as well as the burst firing frequency of Pyra neuron were positively correlated with the dopamine concentrations [8], and the ionic energy consumption of the whole Pyra neuron in MDD group was significantly lower than that in NC group (MDD=0.065 nJ, NC=0.141 nJ) [17].

The simulation results of the actual energy (Figs 4a, 4b and S1d) showed a trend similar to that of the ionic energy [17]: As the dopamine concentration input rose, the actual energy consumption of the Pyra neuron also increased, while the actual energy consumption in NC group was significantly higher than that in MDD group (MDD=0.102 nJ, NC=0.178 nJ). Fig 4b). Unlike MSN, the combined effect of the three external currents was to consume neural energy, rather than supply energy, and this energy component did not differ between MDD and NC groups (MDD= 0.037 nJ, NC=0.037 nJ). Interestingly, the actual energy consumption per spike of the Pyra neuron in MDD group was significantly higher than that in NC group (Fig 4c), indicating that the Pyra neuron in MDD group encoded neural information at a higher cost.

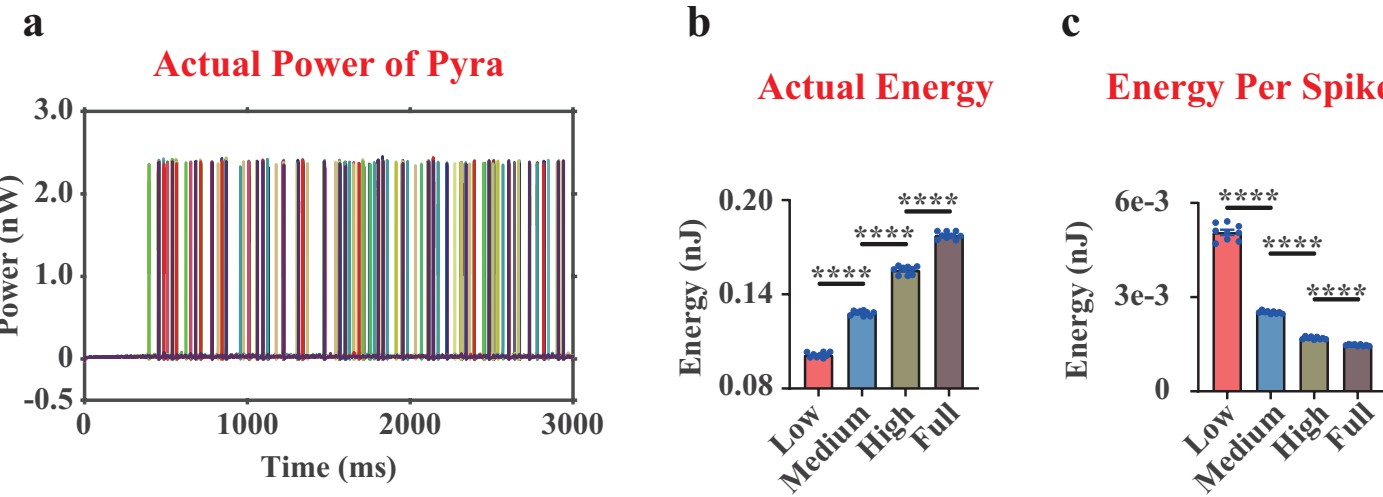

**Fig 4. The actual neural power and actual neural energy results of the whole pyramidal neuron (including one soma, one proximal dendrite and one distal dendrite).** (a), representative curves of the actual neural power of the whole pyramidal neurons (only displaying one replicate under Low dopamine concentration). (b), quantifications of the actual neural energy consumption of the whole pyramidal neuron during 3,000 ms. (c), quantifications of the actual neural energy consumption per spike of the whole pyramidal neuron. Note that there were 20 pyramidal neurons in our model, they were shown in (a) as different colored curves, and each point in (b, c) was the average result of the 20 pyramidal neurons. n=9. ****, p<0.0001.

## Neural energy coding patterns of PV interneurons

Our model included PV interneurons both in the NAc region and the mPFC region [8]. The NAc region had one PV interneuron, whose main role was to provide inhibitory input for the neurons in the NAc region. The mPFC region had three PV interneurons, whose main role was to inhibit the neurons in the mPFC region. Previous studies have found that with the increasing dopamine concentration inputs, there was no significant change in ionic energy consumption of the PV interneuron in the NAc region, but in contrast, the ionic energy consumption of the PV interneuron in the mPFC region significantly increased [17]. This observation suggested that the different network topological structures could lead to distinct neural dynamics and ionic energy consumption patterns even for the same neuronal type [17].

We in this paper investigated the actual energy consumption patterns of the PV interneurons (Fig 5). For PV interneuron in the NAc region, similar to its patterns of ionic energy consumption [17], no significant differences were observed in the actual energy consumption over 3,000 ms or per spike when altering the dopamine concentration inputs (Figs 5a-c and S1b). However, for PV interneurons in the mPFC region, the actual energy consumption patterns (Figs 5e and S1e) were similar to their ionic energy consumption patterns [17], as we also observed a significant increase when increasing the dopamine concentration inputs. In addition, the actual energy consumption per spike of the mPFC PV interneuron in MDD group was significantly higher compared with NC group (Figs 5f and S1e). Interestingly, we found the Pyra neurons and the PV interneurons in the mPFC region showed similar trends both in the actual energy consumption patterns (Figs 4b, 4c, 5b, and 5c) and in the ionic energy consumption patterns [17], suggesting that these two types may have similar energy coding strategy for encoding dopamine activities.

## Decreased actual energy consumption per spike in mPFC CB interneurons under MDD

Our model included one CB interneuron in the NAc region and two in the mPFC region, which had similar network topological structures compared with the PV interneuron type [8]. There was no change in the ionic energy consumption patterns in NAc CB interneuron with the increasing dopamine concentration, but the ionic energy consumption in mPFC CB interneuron showed a significant decrease [17]. The results suggested that the ionic energy consumption patterns of CB interneurons were different from that of PV interneurons due to the different neuronal subtypes [17].

Here, we found that there was no difference in actual energy consumption patterns in CB interneuron in the NAc region (Figs 6a-c and S1c). For the mPFC CB interneurons, although the actual energy consumption over 3,000 ms in MDD group did not differ from that in NC group (Figs 6d, 6e and S1f), the actual energy consumption per spike in MDD group was significantly reduced compared with that in NC group (Fig 6f). Interestingly, this pattern was completely different from our previous findings in the MSN, Pyra neurons and PV interneurons in the mPFC region, which all showed an increase in actual energy consumption per spike with a higher dopamine concentration (Figs 4c and 5f), indicating a different energy coding strategy of mPFC CB neurons.

## Dopamine affects actual energy consumption patterns mainly in the mPFC region

Different dopamine concentration inputs can affect neurodynamics and neural coding patterns of the VTA-NAc-mPFC neural microcircuit [8,16,17]. Previously, we studied the ionic energy consumption patterns in the NAc region and the mPFC region, and the results suggested that

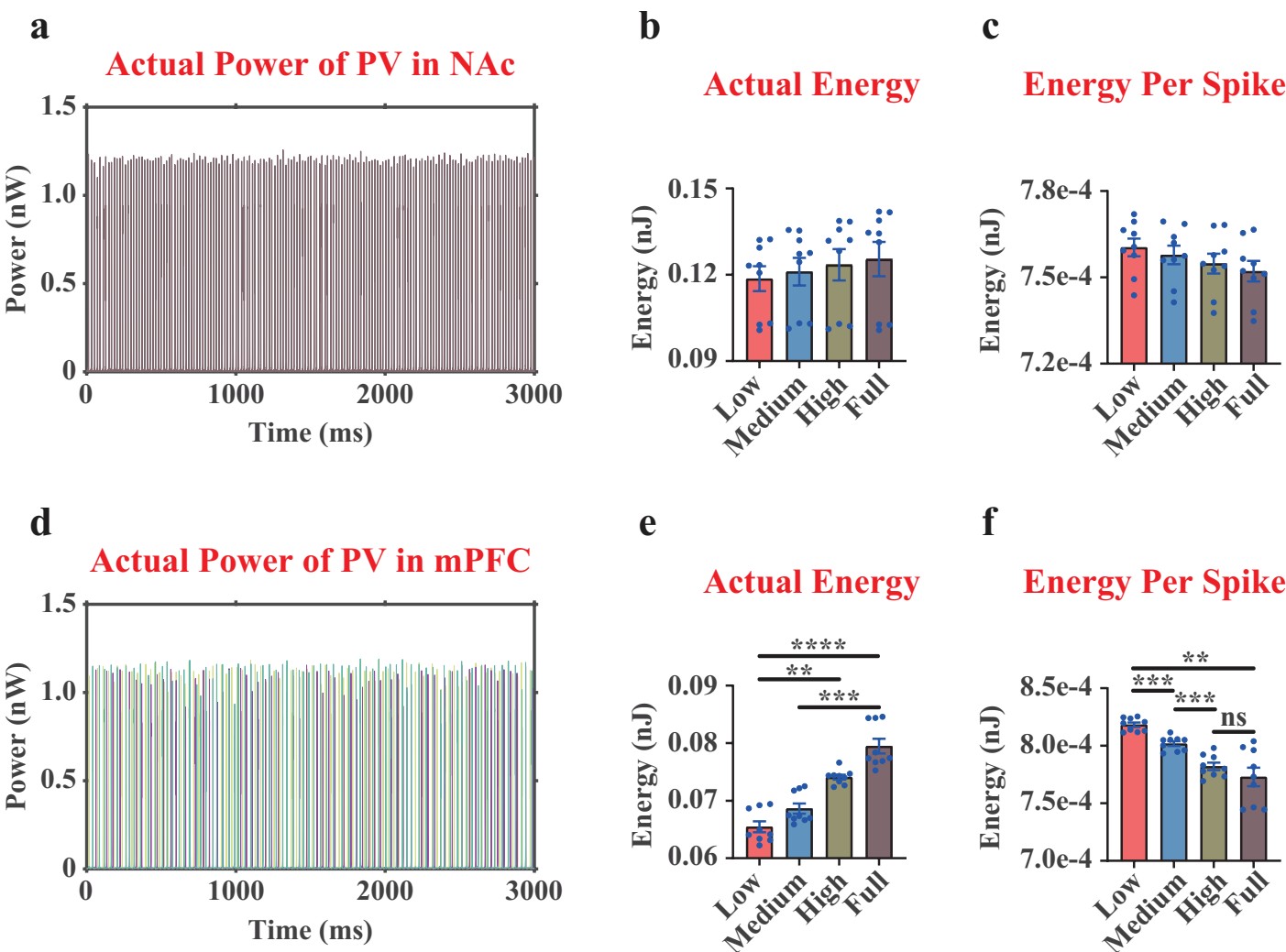

**Fig 5. The actual neural power and actual neural energy results of the PV interneurons in the NAc region and the mPFC region.** (a), representative curves of the actual neural power of the PV interneuron in the NAc region (only displaying one replicate under Low dopamine concentration). (b), quantifications of the actual neural energy consumption of the PV interneuron in the NAc region during 3,000 ms. (c), quantifications of the actual neural energy consumption per spike of the PV interneuron in the NAc region. (d-f), the results of the PV interneurons in the mPFC region. The mPFC region had 3 different PV interneurons, where the curves of different colors represented them in (d). Each point in (e, f) was the average result of the 3 PV interneurons in the mPFC region. n=9. ns, non-significant. **, p<0.01. ***, p<0.001. ****, p<0.0001.

both the two subregions had significantly lower ionic energy consumptions in MDD group (In the NAc region, MDD=0.419 nJ, NC=0.459 nJ; In the mPFC region, MDD=1.556 nJ, NC=3.101 nJ) [17]. However, despite the progress, it remains unclear how dopamine concentration inputs affect the actual energy consumption patterns in the two subregions.

In this paper, strikingly, we found that there was no significant difference in actual energy consumption patterns in the NAc regions between MDD group and NC group (MDD=0.407 nJ, NC=0.407 nJ. Fig 7a-c), suggesting that the external currents of the NAc region supplied more neural energy in NC group (MDD=-0.012 nJ, NC=-0.052 nJ). For the mPFC region, the actual energy consumption in MDD group was statistically lower than that in NC group (MDD=2.316 nJ, NC=3.869 nJ. Fig 7d-f). Further calculations suggested that the external currents of the mPFC region consumed neural energy, and this energy component in MDD group did not differ much from that in NC group (MDD=0.760 nJ, NC=0.768 nJ).

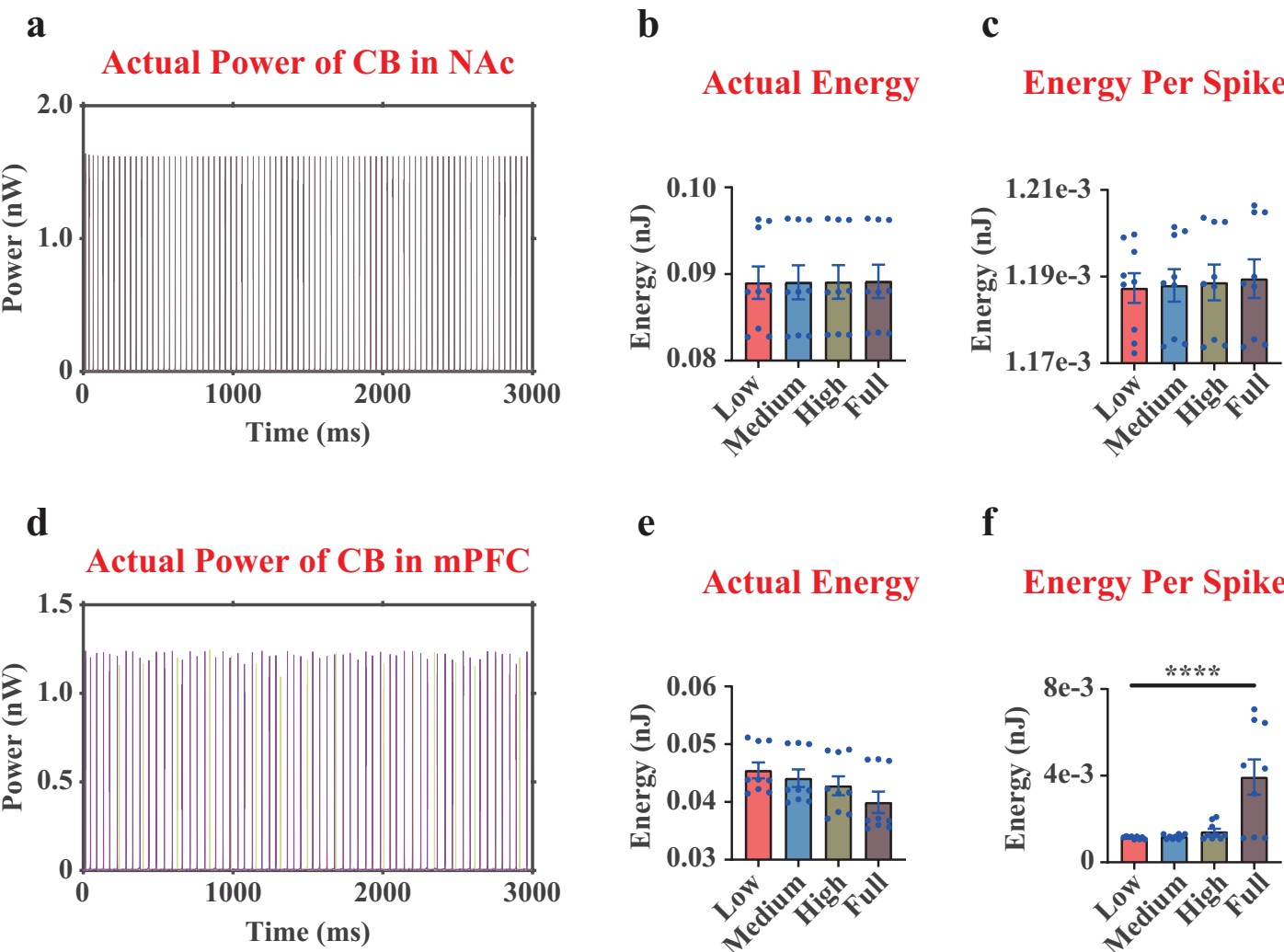

**Fig 6. The actual neural power and actual neural energy results of the CB interneurons in the NAc region and the mPFC region.** (a), representative curves of the actual neural power of the CB interneuron in the NAc region (only displaying one replicate under Low dopamine concentration). (b), quantifications of the actual neural energy consumption of the CB interneuron in the NAc region during 3,000 ms. (c), quantifications of the actual neural energy consumption per spike of the CB interneuron in the NAc region. (d-f), the results of the CB interneurons in the mPFC region. The mPFC region had 2 different CB interneurons, where the curves of different colors represented them in (d). Each point in (e, f) was the average result of the 2 CB interneurons in the mPFC region. n=9. ****, p<0.0001.

We then investigated the actual energy consumption of the whole VTA-NAc-mPFC microcircuit, which showed a significant decrease in actual energy consumption in MDD group (MDD=2.724 nJ, NC=4.276 nJ. Fig 7g-i). This result had a similar pattern with our previous findings of the ionic energy consumption pattern in the whole microcircuit (MDD=1.975 nJ, NC=3.560 nJ) [17]. Accordingly, we reported that the external currents of the whole network in MDD consumed more neural energy than that in NC group (MDD=0.749 nJ, NC=0.716 nJ).

## The P-V correlation between the membrane potential and actual power as a promising indicator for MDD

In the above calculations, we found that based on the different coding theory (traditional neural coding theory and the novel neural energy coding theory), the coding patterns and strategy of our VTA-NAc-mPFC microcircuit could be different. For example, the membrane

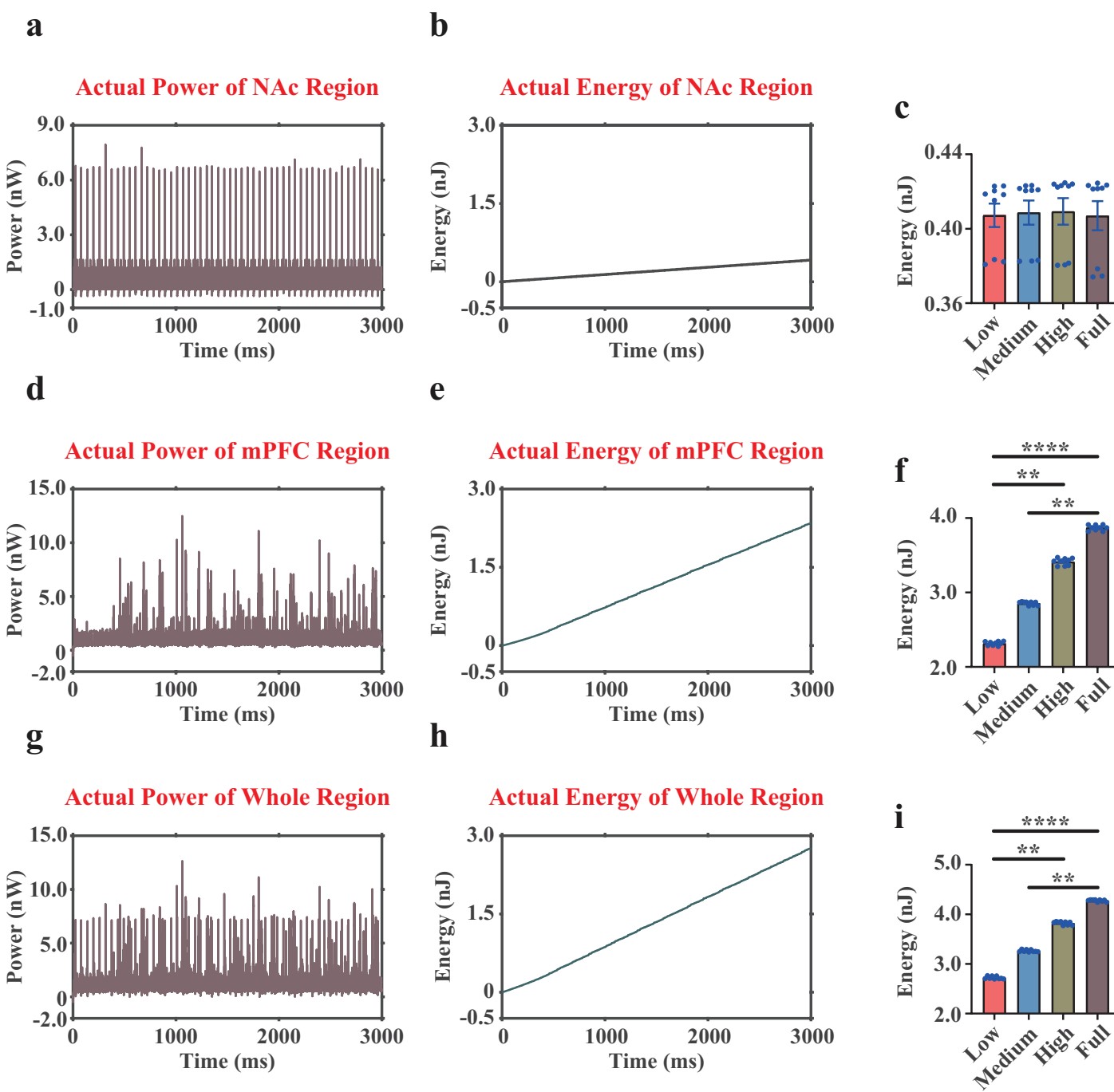

**Fig 7. The actual neural power and actual neural energy results of the NAc region (consisting of one MSN, one PV interneuron, and one CB interneuron), the mPFC region (consisting of 20 Pyra neurons, 3 PV interneuron, and 2 CB interneuron), and the whole microcircuit (28 neurons in total).** (a-b), representative curves of the actual neural power and actual neural energy of the NAc region (only displaying one replicate under Low dopamine concentration). (c), quantifications of the actual neural energy consumption of the NAc region during 3,000 ms. (d-f), the results of the mPFC region. (g-i), the results of the whole microcircuit. n=9. **, $p<0.01$. ****, $p<0.0001$.

potential patterns of the MSN did not differ when altering dopamine concentration, but its neural energy coding patterns had significant differences (Fig 3, and [8,16,17]).

To quantify the difference between the two coding theories in our VTA-NAc-mPFC microcircuit, we introduced the P-V correlation, denoting the Pearson's correlation coefficient between the membrane potential curve and the actual power curve during 3,000 ms. If the P-V correlation was equal to 1, it indicated that the two coding theories were correlated; If the P-V correlation was equal to 0, it indicated that the two coding theories were completely independent. The results exhibited that the P-V correlations for neurons in our VTA-NAc-mPFC microcircuit were between 0.6 and 0.9 (Fig 8), suggesting that the membrane potential curve and the associated neural power curve were not perfectly correlated. We also found that the P-V correlations of MSN, Pyra neurons and CB interneurons in MDD group were significantly lower than those in NC group (Fig 8a, 8d, **and** 8f), indicating that the P-V correlation levels can be used as potential indicators for MDD diagnosis.

## Discussion

Biological experiments at different research levels have shown that the behavioral characteristics of MDD are closely related to the abnormal neural activity patterns and impaired energy metabolism in the brain [34–36], especially in the VTA-NAc-mPFC dopaminergic pathway of the reward neural circuit [3–5,15,21]. However, the relationship between neural activity patterns and neural energy coding patterns of this pathway is still unclear. In this paper, based on our VTA-NAc-mPFC neural microcircuit model [8] and neural energy computational model [17,32], we studied the actual neural energy consumption under different dopamine concentration inputs. We in particular investigated the relationship between the abnormal neural energy coding patterns and neural activity patterns in MDD group, thereby gaining a deeper understanding of the energy coding mechanisms of MDD.

One important view in the field is that MDD is associated with the abnormal glucose metabolism and mitochondrial dysfunction, especially in the mPFC region, which leads to a reduced ability to encode core cognitive functions and a lower neural energy consumption level [15,20–22]. In the current paper, we reported that with the lower dopamine concentration inputs, the actual neural energy consumption of the mPFC region, as well as of the whole microcircuit, both decreased significantly (Fig 7f **and** 7i), which theoretically supported the above view [20–22].

Our paper suggests that different neuronal types exhibit distinct actual energy consumption patterns in response to changes in dopamine concentration levels. This may reflect their varying sensitivity to dopamine within the VTA-NAc-mPFC microcircuit. Especially, the actual energy consumption during 3,000 ms and the actual energy consumption per spike were both changed in MSN (Fig 3b **and** 3c), Pyra neurons (Fig 4b **and** 4c) and PV interneurons in the mPFC region (Fig 5e **and** 5f), suggesting the more sensitive effects of these neurons regulated by dopamine concentration. In addition, MSN and Pyra neurons exhibited different patterns of actual energy consumption (Figs 3b and 4b), which may be explained by their distinct dopamine receptor types, since the MSN was inhibitory D2-type and Pyra was excitatory D1-type in our model [8]. We also observed different energy consumption patterns between PV and CB interneurons (Figs 5 and 6). Theoretically, PV and CB interneurons in the mPFC region has different types of ion channels and synaptic current kinetics (see S1 File), which together contributed to distinct neurodynamics [8] and neural energy coding patterns across interneuron subtypes. From a biological perspective, the fact that PV interneurons in the mPFC region exhibited greater changes in energy consumption under varying dopamine input suggested that PV interneurons were sensitive to dopamine modulation, which was in line with previous experimental observation [37].

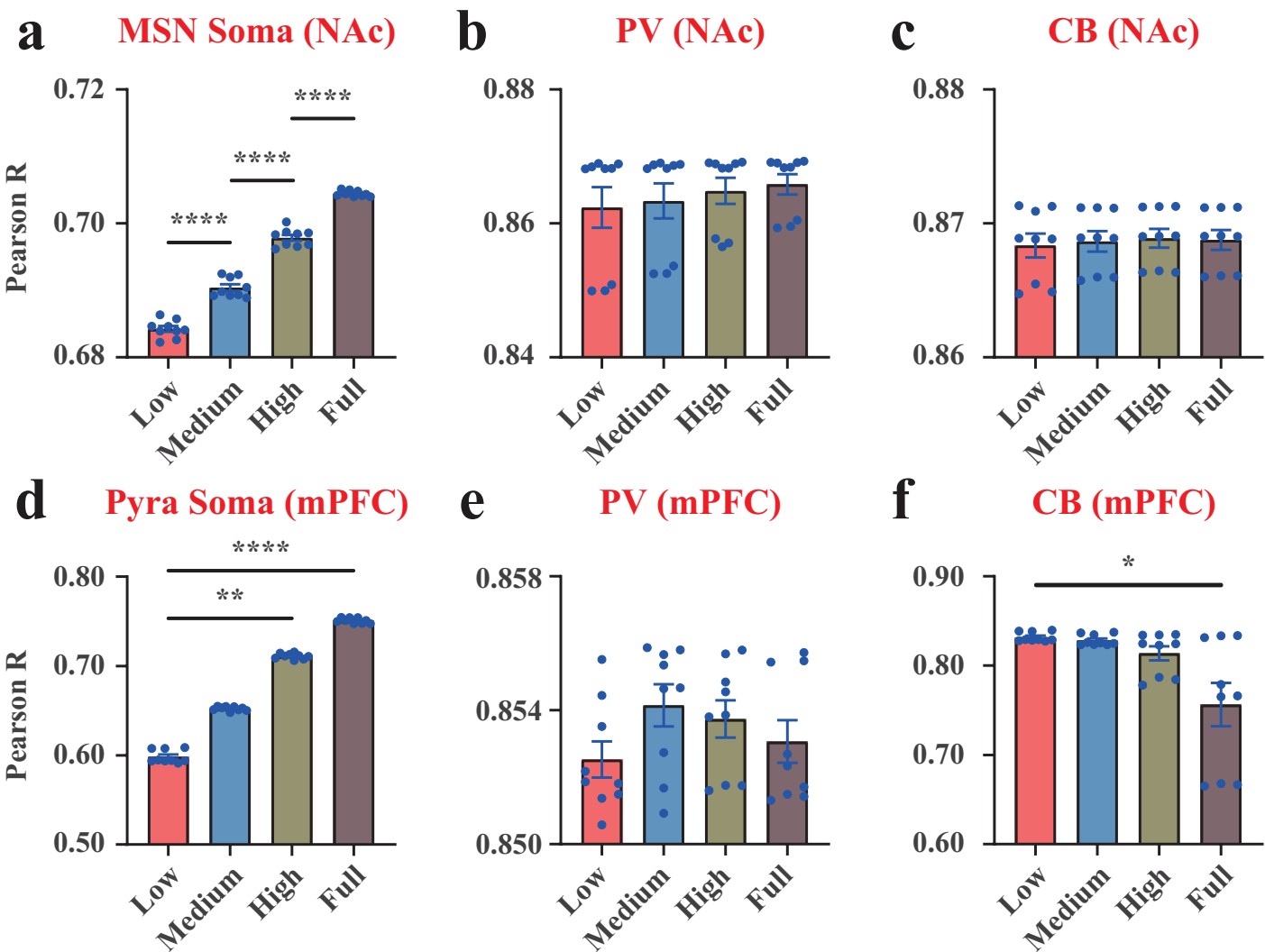

**Fig 8. The Pearson's correlation coefficients between the membrane potential curve and the actual neural power curve (P-V correlations).** (a), results of the MSN soma in the NAc region. (b), results of the PV interneuron in the NAc region. (c), results of the CB interneuron in the NAc region. (d), results of the Pyra soma in the mPFC region. (e), results of the PV interneuron in the mPFC region. (f), results of the CB interneuron in the mPFC region. Each point in (d) was the average result of the 20 Pyra neurons in the mPFC region. Each point in (e) was the average result of the 3 PV interneurons in the mPFC region. Each point in (f) was the average result of the 2 CB interneurons in the mPFC region. n=9. *, p<0.05. **, p<0.01. ****, p<0.0001.

The investigation of actual energy consumption patterns was a complement to our past findings on ionic neural energy [17]. Since the ionic neural energy denoted the theoretical energy required to move the ions across the cell membrane through ion channels, and the actual energy consumption measured the actual neural energy consumption of a neuron in the microcircuit, they together reflected the neural energy coding patterns of the VTA-NAc-mPFC microcircuit.

In our previous study [17], we demonstrated reduced ionic neural energy consumption in the VTA-NAc-mPFC circuit of the MDD group compared to the NC group, and the current study further observed decreased actual neural energy consumption in the same circuit of the MDD group. Interestingly, we found that for the MSN, although its ionic energy consumption in MDD group was lower than that in NC group [17], its actual energy consumption in MDD

group was conversely higher than that in NC group (Fig 3b). Further calculations suggested that the synaptic currents of MSN supplied more neural energy in NC group (Fig 3d), which was caused by an increase in energy supplied by EPSCs (Fig 3e). The more detailed classification of neural energy component can allow us to get a more nuanced understanding of energy coding patterns.

In conclusion, our paper suggested that some patterns of neural energy consumption were significantly affected by the dopamine concentration inputs, such as the actual energy consumption (Figs 3b and 4b), the energy efficiency (Figs 3c and 4c) and the P-V correlations (Fig 8a and 8d) of MSN and Pyra neurons, as well as the actual energy consumption patterns of the mPFC region (Fig 7f) and the whole neural microcircuit (Fig 7i). Upon these specific neural energy coding patterns, we proposed that neural energy can be used as a potential indicator for MDD diagnosis.

In addition, based on the neural coding theory [9] and neural energy coding theory [18], we have established a new pipeline for studying MDD [8,16,17]. By quantifying the P-V correlations between the two coding theories in our VTA-NAc-mPFC microcircuit (Fig 8), we proposed that the joint application of the two theories will be superior to the application of any single theory, and this joint application could help discover new mechanisms in neurocircuits of MDD. The simulation results also supported this perspective, e.g., there was no difference in the membrane potential coding patterns in MSN under different dopamine concentration inputs [8,16], but differences in neural energy coding pattern were observed in Fig 3 and [17].

Despite these contributions, we note that the current study has a limitation. Our neurodynamic VTA-NAc-mPFC model is a simplified neural network model in which we have reduced the complexity of neuron types (e.g., excluding cholinergic neurons [4]) and synaptic composition (e.g., not accounting for dopaminergic neurons that utilize glutamate as a co-transmitter [38]). In addition, although our neurodynamic VTA-NAc-mPFC model successfully simulates membrane potentials, ion channel currents, synaptic currents, and dopamine concentrations, and the findings of neural energy consumption patterns in this paper were in line with the biological principles, it remains a theoretical framework derived from experimental data. As noted in our previous studies [8,17], validating this model experimentally remains challenging due to current technological limitations. Moreover, unlike conventional approaches that estimate neural energy based on ATP consumption and the sodium-potassium pump [10,39,40], our model based on REF [32] directly computed the electrical power of the circuit based on the H-H equations, and we did not compare these different methods. Besides, our current findings are based on a microcircuit model, and we will further consider its application to large-scale neural networks in the future. Nonetheless, as a preliminary study in this field, we therefore plan to explore other experimental or theoretical methods that can support our current findings.

## Supporting Information

**S1 File. This file shows the neurodynamical modeling and neural energy computation methods for the VTA-NAc-mPFC neural microcircuit.**
(PDF)

**S1 Fig. The regression lines between dopamine concentration and actual energy or energy per spike in different neuronal types.** Since the dopamine concentration was set as a uniform distribution within a range, the midpoint of the interval was taken as the dopamine concentration for plotting (Low – 0.125, Medium – 0.375, High – 0.625, Full – 0.875). $p$ values are not corrected here.
(PDF)

**S1 Data. Data and statistics based on the neural energy simulation results.**
(ZIP)

## Author contributions

**Conceptualization:** Yuanxi Li, Bing Zhang, Rubin Wang.

**Data curation:** Yuanxi Li, Jinqi Liu.

**Formal analysis:** Yuanxi Li, Bing Zhang, Jinqi Liu.

**Funding acquisition:** Bing Zhang, Rubin Wang.

**Investigation:** Yuanxi Li.

**Methodology:** Yuanxi Li, Bing Zhang, Rubin Wang.

**Software:** Yuanxi Li, Jinqi Liu.

**Validation:** Yuanxi Li.

**Visualization:** Yuanxi Li, Jinqi Liu.

**Writing – original draft:** Yuanxi Li.

**Writing – review & editing:** Yuanxi Li, Bing Zhang, Rubin Wang.

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
