## [Decision Letter · Decision Letter 0]

30 Dec 2024

PCOMPBIOL-D-24-01550

Neural energy coding patterns of dopaminergic neural microcircuit and its impairment in major depressive disorder: A computational study

PLOS Computational Biology

Dear Dr. Li,

Thank you for submitting your manuscript to PLOS Computational Biology. After careful consideration, we feel that it has merit but does not fully meet PLOS Computational Biology's publication criteria as it currently stands. Therefore, we invite you to submit a revised version of the manuscript that addresses the points raised during the review process.

Please submit your revised manuscript within 60 days Mar 01 2025 11:59PM. If you will need more time than this to complete your revisions, please reply to this message or contact the journal office at ploscompbiol@plos.org. Please include the following items when submitting your revised manuscript:

We look forward to receiving your revised manuscript.

Kind regards,

Jan Karbowski

Guest Editor

PLOS Computational Biology

Hugues Berry

Section Editor

PLOS Computational Biology

**Additional Editor Comments :**

It is surprising, given that they consider neuronal energy, that the authors do not cite the classic paper by Attwell and Laughlin (2001) about energy budget in the brain (J. Cereb. Blood Flow Metabol. 2001). This paper should be mentioned and cited in the text. Also there is another, similar paper in topic, which should be cited in the text - Yu et al "Cerebral Cortex" 2022 ("A 3D atlas of functional....").

The energy model in the paper (Eqs. 2 and 3) is somewhat abstract. In particular, it is not clear what is the biophysical sense of the total energy H_all? It should be explained that H_all relates to the ATP used by Na-K pumps that maintain concentration gradients of different ions. For this more biological approach, see J. Karbowski - J. Comput Neurosci. 2009 ("Thermodynamic constraints on neural dimensions .... "), and J. Karbowski - PLOS One 2012 ("Approximate invariance of metabolic energy...."). The authors should include these considerations in their paper.

**Journal Requirements:**

At this stage, the following Authors/Authors require contributions: Yuanxi Li, Bing Zhang, Jinqi Liu, and Rubin Wang. Please ensure that the full contributions of each author are acknowledged in the "Add/Edit/Remove Authors" section of our submission form.

Potential Copyright Issues:

i) Figures 1, Thank you for stating that "The figure was reproduced from Li et al. [8, and 15] ." Please provide written permission from the copyright holder to publish this under our CC-BY 4.0 license, or remove the figure / replace the image. Please note we do not recommend using standard request forms available on Publishers' websites, as they grant single use rather than republication under an open access license.

6) In the online submission form, you indicated that The MATLAB codes that support the findings of this study are available at GITHUB (https://github.com/Yuanxi-Li/MDD-NeuralEnergy) upon reasonable request after acceptance. All PLOS journals now require all data underlying the findings described in their manuscript to be freely available to other researchers, either

1. In a public repository

2. Within the manuscript itself

3. Uploaded as supplementary information.

7) We found that this link "(https://github.com/Yuanxi-Li/MDD-NeuralEnergy)" reaches 404 error page. Please amend it to a new link.

8) Please amend your detailed Financial Disclosure statement. This is published with the article. It must therefore be completed in full sentences and contain the exact wording you wish to be published.

2) State what role the funders took in the study. If the funders had no role in your study, please state: "The funders had no role in study design, data collection and analysis, decision to publish, or preparation of the manuscript.".

**Reviewers' comments:**

Reviewer's Responses to Questions

Reviewer #1: Yuanxi Li et al (PCOMPBIOL-D-24-01550)

The paper entitled “Neural energy coding patterns of dopaminergic neural microcircuit and its impairment in major depressive disorder: A computational study” by Li et al is an advance on prior work from this group which developed neural network modeling of the VTA-NAc-mPFC neural microcircuit using the basic Hodgkin-Huxley (H-H) model but allowing it to be linked to neurotransmitter level, which in their case is dopamine. This advancement thereby allows neurotransmitter level to be linked to the H-H estimates which are largely based on ionic currents. The current work is an extension of the prior model to calculate neural energy based on the VTA-NAc-mPFC neural microcircuit model, specifically instantaneous neural power corresponding, respectively, to Equations (1) and (2) in their paper. They studied the neural energy consumption under different dopamine inputs and related their neurotransmitter levels to energy consumed and energy/spike. In the current work, the ‘Low’ dopamine was considered as the MDD group, whereas the ‘Full’ dopamine was considered as the NC group. Quite sadly, no specific conclusion was made about MDD vs. NC per se. Instead, very general conclusions were made. While there are no major errors in the work, as the authors themselves state, the work is purely computational and thus it is difficult to assess how their work could be useful in the knowledge of neurotransmitter levels vs. neuroenergetics in health and disease. Please note that since both of these metrics (neurotransmitter vs. neuroenergetics) can be measured in vivo, it is important to reveal how their work is relevant in MDD, but also other psychiatric disorders.

1. While it is stated that Equations (1) and (2) respectively reflect the prior and current work, it is important to show how ionic currents in the H-H equations are related to levels of neurotransmitter. Some clear statements in the Introduction and Discussion, with supporting supplementary information, about the relation between ionic currents vs. membrane potential (which is very well known from prior work on glutamatergic and GABAergic systems) and neurotransmitter level vs. membrane potential (which is less well known and is the novelty of their work). It is crucial to identify these relations that are widely applicable across all neurotransmitter systems.

2. It is assumed that most readers know distributions of different neurotransmitters in VTA, NAc, mPFC, etc, or do the authors assume the ratio of different neurotransmitters to be the same across these regions? Both can be true. At present the authors don’t speak about different neurotransmitter levels in any region. Please let the readers know in Methods using absolute units (mM) how major neurotransmitters (glutamate, GABA, dopamine, serotonin, etc) vary across these regions. This information should then be showed in Methods as varying model equations for each region, as one of their conclusions is that parvalbumin interneuron (PV) and calbindin interneuron (CB) in NAc have different behaviors. Why?

3. The authors state that "Numerous experiments have found that the behavioral characteristics of major depressive disorder (MDD) animals are usually associated with abnormal neural activity patterns and brain energy metabolism." Please clarify by which methods these energetic values were measured, or are they just referring to computational predictions? Unclear.

4. The authors state that they "not only uncovered the neural energy coding patterns for the VTA-NAc-mPFC neural microcircuit, but also presented a novel pipeline for the study of MDD based on the neural coding theory and neural energy coding theory." The former is more important than the latter. So please compare for each region and the circuit together for clarifying this statement further. The Abstract should not have vague statements as nothing conclusive is stated about MDD although it’s in the title.

5. Although a recent discovery, it has been shown that dopaminergic neurons allow co-transmission of glutamate across phyla, from flies to humans. How does their conclusion change if certain fraction of dopaminergic neurons inherently co-transmit glutamate? This should at least be a discussion point of weaknesses.

6. In MDD and other psychiatric studies, the attention seems to be on the unique cellular architecture of dopaminergic neurons, specifically their large and complex arborization of axons, which puts these neurons under a heavy energy budget limit and thus makes them particularly susceptible to factors that contribute to subsequent cell death. However, if these cells are only 1% or 5% across regions, for their loss to have energetic impact on that region depends on their energy budget. The field of energy budget for glutamatergic and GABAergic systems, across phyla and in the human brain, have been well developed. How do the energy budgets of these neuronal systems vary?

7. The section in Discussion that compares the energetics of different regions, it is necessary to convert the nJ units to mircomole/g/min – which is the main unit for metabolic measures by in vivo methods. This is important to understand the impact of regional differences regardless of their neurotransmitter levels.

8. Please avoid jargon before they are defined, for example, Pyra? I understand it’s pyramidal neurons etc. So please have a table for all abbreviations used with units converted for comparisons.

Minor comments:

- Fig 4 (glutamatergic) and Fig 5 (GABAergic) need to do comparisons of B and C figures for slopes and exponent etc

- Comparison of Fig 5/6 (both inhibitory neuronal systems) need clarification as to why CB is unresponsive

- plot Fig 7b,e,h in the same scale so that regional results can be appreciated

Reviewer #2: The quite well-written paper presents a biophysical model for a core circuit that includes the ventral tegmental area (VTA), the

nucleus accumbens (NAc), and the medial prefrontal cortex, which is based on data from rodent experiments. The model is based on modified Hodgkin-Huxley equations for a total of 28 neurons, taking into account the specific properties of different relevant channels, and the Calcium dynamics, and also making specific assumptions about a highly simplified morphology of the

neurons. The paper studies the behavior of this simulated circuit in major depressive disorder (MDD) by modeling variations changes in the neural dynamics that are dependent of changes of the dopamine concentration. The paper compares two ways of analysis, a classical analysis of the neural activity patterns and an analysis that investigates the electrical energy in different parts of the neural circuit. Both ways of analysis result partly in different dependencies of the behavior on the dopamine

concentration, so that the neural energy analysis is proposed as an alternative way for analyzing the behavior of such neural structures, and maybe also a method to develop new clinical markers for system dysfunction.

Major points:

---------------

The paper presents a solid biophysical simulation study that contrasts neural activity patterns and an analysis in terms of the electrical energy of neural signals. Overall, the results seem convincing, and the approach is original. The study raises the following questions, which do not become entirely clear and that should be addressed prior to publication.

1) The paper proposes neural energy analysis as alternative way of analyzing disease related changes. While it is demonstrated that by simulation the model demonstrates specific changes in energy measures, which partly deviate from neural activity changes, the question emerges how such energy quantities can be measured directly in such circuits. This would make them most suitable as alternative clinical markers.

2) The modeling study is based on 9 repeated simulation of a model that contains at least 9 randomly chosen parameters. Is this number of repetitions really enough for obtaining stable results? This should be demonstrated or at least an argument should be given, why this is enough.

3) The model makes very specific assumptions about a simplification of the anatomy of the underlying model and assumes in total only 28 neurons. Why was in particular this number of neurons chosen ? How sensitive are the results with respect to these assumptions. Ideally, some of these parameters or assumptions could be changed, showing that the key results hold even for a variation of some of these parameters.

Minor points:

-------------

(page numbers counted from page after the abstract (= p. 1)

* p.2, l. 21ff: What is the justification for the 28 neuron model and the relevant simplifications?

* p. 7, last paragraph: I am wondering if the Pearson correlation coefficient is the right statistical measure for the relationship between the variables, since they might not be normally distributed. Otherwise, a Spearman correlation might be more appropriate. Also is seems that the correlations themselves apparently were not tested for significance.

* Are the 'Pyra neurons' pyramidal cells?

* Can there be an argument given why this assumption of uniform distributions of the

different model parameters makes sense ? Same applies to the assumption of uniformly distributed

connection strengths, which should have a strong influence on the network dynamics

(omega parameters in formula S-12).

* In the Supplement the derivation of the conductance of the compartment current using the Rall model

(equation S-7) did not become clear to me. Either here a paper should be cited where this exact formula

is derived, or an explicit explanation should be given, what exactly the underlying derivation

is based on (length currents within the neurites; boundary conditions for the cable part etc.)

* Supplement, section 3, last paragraph: Were these assumptions which conductances change with dopamine based on experimental data? On which one?

**Have the authors made all data and (if applicable) computational code underlying the findings in their manuscript fully available?**

Reviewer #1: Yes

Reviewer #2: Yes

PLOS authors have the option to publish the peer review history of their article (what does this mean? ). If published, this will include your full peer review and any attached files.

**Do you want your identity to be public for this peer review?** For information about this choice, including consent withdrawal, please see our Privacy Policy .

Reviewer #1: No

Reviewer #2: **Yes: ** Martin A. Giese

**Figure resubmission:**
---

## [Editor Report · Decision Letter 1]

13 Mar 2025

Dear Dr. Li,

We are pleased to inform you that your manuscript 'Neural energy coding patterns of dopaminergic neural microcircuit and its impairment in major depressive disorder: A computational study' has been provisionally accepted for publication in PLOS Computational Biology.

Best regards,

Jan Karbowski

Guest Editor

PLOS Computational Biology

Hugues Berry

Section Editor

PLOS Computational Biology

---

## [Editor Report · Acceptance letter]

PCOMPBIOL-D-24-01550R1

Neural energy coding patterns of dopaminergic neural microcircuit and its impairment in major depressive disorder: A computational study

Dear Dr Li,

I am pleased to inform you that your manuscript has been formally accepted for publication in PLOS Computational Biology. Your manuscript is now with our production department and you will be notified of the publication date in due course.

With kind regards,

Anita Estes
